# Review of the Distribution and Influence of Antibiotic Resistance Genes in Ballast Water

**Jiaqi Guo, Bo Jiang, Sumita, Chengzhang Wu, Yunshu Zhang * and Cong Li ***

School of Environment and Architecture, University of Shanghai for Science and Technology, Shanghai 200093, China

* Correspondence: zhangys@usst.edu.cn (Y.Z.); congil@aliyun.com (C.L.); Tel.: +86-021-5527-5979 (Y.Z.); +86-176-4503-2470 (Y.Z.)

**Abstract:** The misuse of antibiotics causes antibiotic resistance genes (ARGs) in bacteria to be gradually enriched by environmental selection, resulting in increased tolerance and resistance in bacteria to antibiotics. Ballast water is a mobile carrier for the global transfer of bacteria and genes, thus posing a certain risk of ARGs spreading into the global ocean. Therefore, it is important to investigate the current status of ARGs in ballast water, as well as control the abundance of ARGs. Herein, we attempt to comprehensively summarize the distribution and abundance of ARGs in ballast water from different sea areas and analyze the influencing factors (such as physical factors, chemical factors, temperature, pH, etc.) on the distribution of ARGs. Furthermore, we seek to review the changes in ARGs after differential disinfection technology treatment in ballast water (including chlorination, ultraviolet, ozone, and free radical technology), especially the enhancing effect of subinhibitory concentrations of disinfectants on ARGs transfer. Overall, we believe this review can serve as a guide for future researchers to establish a more reasonable standard of ballast water discharge that considers the pollution of ARGs and provide new insight into the risk of vertical and horizontal ARG transfer in ballast water after disinfection.

**Keywords:** ballast water; antibiotic resistance genes; disinfection; transfer





## 1. Introduction

Ballast water is the water that fills the bottom and side tanks of a ship to keep the hull balanced and accounts for approximately 40% of the cargo transportation volume [1]. According to the estimates of the International Maritime Organization (IMO), each year, up to 12 billion tons of ballast water are transshipped around the world, and every year more than 80,000 ships transport ballast water around the world [2]. The transfer and discharge of ballast water from ships is currently the main pathway leading to the invasion of alien organisms through offshore waters (about 29% of total biological invasions) [3], and it is estimated that approximately 3000 species are transferred through ship ballast water every day [4]. A large number of potentially pathogenic microorganisms have been detected in river estuaries, ports, and bays, posing potential harm to aquatic organisms and human health [5]. Therefore, ballast water is a mobile carrier for the global transfer of bacteria and genes, which has been attracting much attention in recent years [6].

Recently, various residual antibiotics found in the marine environment have become a focus of public concern [7–9]. The high concentrations of antibiotics in the marine environment are mainly derived from chemical production, breeding, aquaculture, and medical care [10]. Additionally, some kinds of antibiotics are resistant to degradation, resulting in their environmental accumulation in marine environments. Antibiotics at high concentrations can threaten the biodiversity and function of adjacent aquatic ecosystems and may cause bacteria to produce antibiotic resistance genes (ARGs) and threaten human health [11]. If antibiotics are allowed to remain in the marine environment, they will exert selective pressure on bacteria, inducing the enrichment of ARGs. ARGs can be horizontally

transferred between different strains through mobile genetic elements (MGEs) [12,13], thereby enabling more microorganisms to acquire antibiotic resistance. Gene transfer from ARGs can allow antibiotic-resistant pathogenic bacteria to emerge. It also endangers people's health and disturbs the ecosystem's balance [14].

The aquatic ecosystem is a huge natural repository of ARGs. Recently, with frequent human activities, the increase in wastewater discharge, and the development of marine aquaculture activities, the abundance of ARGs in the marine environment has gradually increased [10,15]. Therefore, antibiotics from agriculture and human waste are often found in ballast water near the coast [16,17]. Because medical treatment and animal husbandry use a lot of antibiotics, medical wastewater and aquaculture wastewater are considered to be the main source of antibiotic-resistant genes [18,19]. Wastewater and domestic sewage containing antibiotics or resistance genes will be directly or indirectly discharged into rivers [6], lakes [20], and oceans. Therefore, the sea will become the final place for the influx of resistance genes [21]. There are three possible sources of ARGs in coastal waters. The first source is the direct input from land sources, and surface runoff carrying ARGs enters the coastal waters [22]. For example, effluent from sewage treatment plants [23–25]. The second source is the ARGs produced by the selective pressure of antibiotics or other poisons used in water aquaculture [26]. The third source is the inherent background value of those antibiotic resistance genes evolved by microorganisms in the marine environment to resist other populations that can secrete antibiotics [27].

Ballast operations are usually carried out in offshore waters [28]. Therefore, seawater containing various bacteria [29–32], ARGs [33,34], and antibiotic residues [6] can be pumped into the ship's ballast tank. When antibiotic-resistant bacteria in ballast water are gradually enriched, the transfer of antibiotic-resistant genes between microbial communities occurs [28]. It has been proven that there are more ARGs in ship ballast water than in samples taken from the nearby ocean [5,35,36]. In addition, ARGs were found in many samples of ballast water from different sea areas, which shows that ballast water can spread antibiotic resistance [21]. Therefore, it is necessary to control the abundance of ARGs in ballast water.

When minimizing the hazards from ballast water discharge, ballast water must be discharged after treatment. At present, relevant studies have shown that adsorption, activated-sludge technology [37–39], and constructed wetlands [40] have been used to deal with ARGs pollution. These methods may cause secondary pollution problems. In these treatment methods, ARGs are mainly transferred rather than completely degraded. This paper summarized the common disinfection methods of antibiotic resistance genes and summarized their advantages and disadvantages. The common and traditional disinfection processes mainly include ultraviolet disinfection [41] and chlorination disinfection [42]. Free radical disinfection is currently a research hotspot, including the Fenton reaction [43], photocatalysis [44,45], and photoelectric catalysis [46], which show great advantages in removing ARGs.

In summary, the discharge of ballast water accelerates the spread and diffusion of ARGs in the marine environment, with each pollution source and carrier diffusing exponentially [47]. It is necessary to summarize the distribution and abundance of ARGs in ballast water. In addition, there are few reviews on the abundance changes of ARGs and the transfer risk of ARGs in ballast water after disinfection. Therefore, the main contents of this review mainly include (1) introducing the distribution and abundance of ARGs in ballast water from different sea areas and different sources; (2) investigating the abundance of resistance genes in ballast water after disinfection, as well as the vertical and horizontal transfer of ARGs under nonantibiotic stress mediated by different disinfection methods.

## 2. Methodology

### 2.1. Methods

In order to understand the distribution and impact of antibiotic resistance genes in ballast water, we reviewed the most relevant studies on the use of antibiotics, the generation

of antibiotic-resistant bacteria in water, the biological invasion caused by transmission and ballast water transportation, and the transmission of antibiotic resistance genes. This paper combs the research on the distribution and transfer of antibiotic resistance genes in the past 25 years from several aspects, such as the issuing agency, the author's cooperation network, keyword and research evolution, digging the frontier hot spots in this field, looking forward to identify the future research trends in this field, and provide a certain reference for the related research and time development to solve the harm of antibiotic resistance genes.

*2.2. Identifification of Records*

The scope of this paper is to review the distribution and impact of antibiotic resistance genes in ballast water, especially the papers that collect the distribution of antibiotic resistance genes in ballast water transportation, the impact of antibiotic resistance gene transfer on the environment, and the disinfection methods of antibiotic resistance genes. To ensure the quality of the comments, we collected peer-reviewed journal articles and research reports indexed by Science Direct and Web of Science databases. In collecting these references, we used any search term in the title, abstract, and keywords to identify the relevant research. Among them, the keywords "ballast water", "antibiotic resistance gene", "antibiotic-resistant bacteria", "resistance gene transfer", "resistance gene removal", and "ballast water disinfection" were used to determine the reference library. The time range of the references in this review span 1998 to 2022.

*2.3. Eligibility Criteria*

The purpose of this review is to include searchable studies on the distribution of antibiotic resistance genes in ballast water and the disinfection of resistance genes during transportation. The main research that met the conditions for this review required the following criteria: (1) the distribution of resistance genes in ballast water was analyzed; (2) the effect of the disinfection technology on the inactivation of the resistant genes was analyzed; (3) the effects of the minimum inhibitory concentration on the horizontal transfer of resistant genes were analyzed.

## 3. Bacteria in Ship Ballast Water

At the Diplomatic Conference of the International Maritime Organization in 2004, the "International Convention on the Control and Management of Ships Ballast Water and Its Sediments" was passed. This is also called the "Ballast Water Convention". The ballast water convention mainly has five chapters (chapters A–E). All ships subject to the ballast water convention have fully entered the stage of implementing ballast water management, which is the second regulation in Chapter D, namely, the D-2 standard [48] (Table 1). At present, people's attention to ship ballast water discharge is focused on this standard.

**Table 1.** Microorganism discharge limitations based on the IMO D-2 Standard (IMO, 2004) [48].

| | | Discharge Limitation Colony form Ingunit (cfu) |
|---|---|---|
| Indicator microbes | Toxicogenic *Vibrio cholerae* *Escherichia coli* (*E.coli*) Intestinal *Enterococci* | <1 cfu per 100 mL <250 cfu per 100 mL <100 cfu per 100 mL |
| Size of microorganism | $\geq$50 μm $\geq$10 μm and <50 μm | <10 viable organisms per m$^3$ <10 viable organisms per mL |

Even though disinfection may remove the majority of the bacteria in ballast water, there are still many bacteria in the water that can spread rapidly to new maritime environments. According to reports, pathogenic germs and other types of bacteria have been discovered in ballast water, mostly in Chinese ports [35]. In addition, bacteria may also affect the sea area of the port with the ship's ballast water after long-distance migration [49]. For

example, Lv et al. [50] used 16 S rRNA technology to analyze sediment samples from the ballast tanks of nine ocean-going ships in Jiangyin, China, and detected 44 pathogenic bacteria from the sediments, among which *Pseudomonas alkalogenes*, *Enterococcus Heidegger*, *Shigella*, and *Bacillus anthracis* were the most abundant, accounting for 64.5% of the total pathogenic bacterial sequences. Yang et al. [51] found nine kinds of pathogenic bacteria in the ballast water of the Yangshan Port, including *Pseudoalteromonas piscicida*, *Rhodococcus erythropolis*, *Vibrio* sp., *Bacillus aquimaris*, *Vibrio alginolyticus*, etc. Altug et al. [6] investigated the ballast water from ships sailing to Turkey in the Marmara Sea and found 38 species, including 27 pathogens, which shows that ballast water could be harmful. Salleh et al. [52] investigated the ballast water from eight different sources, and 33 potential pathogens were detected from all of the ballast water samples. Among them, *Pseudomonas* spp., *Tenacibaculum* spp., *Flavobacterium* spp., *Halomonas* spp., and *Acinetobacter junii* are the main potential pathogens.

Furthermore, Ng et al. [53] analyzed the ballast water of six cabins in a Singapore port and found a variety of *vibrios* that showed antibiotic resistance. Rivera et al. [54] studied ballast water in Brazil and found the *Vibrio cholerae* O1-producing virus in the ballast water samples. Invasive *Vibrio cholerae* strains can be further transported to other regions by ships, which will bring harm to other regions. Dobbs et al. [55] also proved that *Vibrio cholerae* in ship ballast water gradually acquired resistance to antibiotics, which indicated that *Vibrio cholerae* isolates could transfer ARGs to other *Vibrio cholerae* strains by horizontal transfer. This phenomenon may be enhanced in the ballast tank [56]. Therefore, ballast water is not only a carrier of bacteria for global spreading, but it is also a source of genes.

The spread of bacteria, especially pathogenic bacteria carried by ballast water, not only threatens the local marine environment and aquatic fisheries but also threatens human life [55]. On the other hand, the transfer of drug-resistance genes to pathogenic bacteria will gradually enhance the threat of pathogenic bacteria [47]. In recent years, the existing microbial detection basis for ballast water have tended to be standardized; however, ARGs are an emerging pollutant that have been neglected in the detection of ballast water. Therefore, the discharge index of ballast water needs to be changed, and the management of ballast water discharge needs to be investigated [57].

## 4. Resistance Genes

### 4.1. Antibiotics and Antibiotic Resistance Genes

Antibiotics, which are important growth inhibitors to prevent bacterial diseases and stimulate growth in aquaculture, are often used as disease-prevention drugs and feed additives in aquaculture [10]. At present, the commonly used antibiotics with a high detection frequency and detection abundance in the environment mainly include the following categories (Table 2): tetracyclines, sulfonamides, quinolones, β-lactams, aminoglycosides, macrolides class, and amide alcohols (also known as chloramphenicol antibiotics) [9]. However, because of the abuse of antibiotics, bacteria suffer from sustained antibiotic stress, which induces the enrichment of ARGs in bacteria. Recently, antibiotic resistance has become a hot issue of global concern [12]. Antibiotic-resistant bacteria always show multiple antibiotic resistance (MARs), leading to the emergence of multiantibiotic-resistant bacteria and even "superbugs" that can tolerate most antibiotics [48]. Furthermore, ARGs will accumulate, diffuse, and be transferred into the environment.

**Table 2.** Common types of antibiotics and representative drugs [58].

| Category | Effect | Representative Drug |
| --- | --- | --- |
| Tetracyclines | It is an antibiotic with a broad spectrum that destroys a variety of bacteria. Its primary function is to inhibit peptide chain growth and bacterial protein synthesis. | Tetracycline, oxytetracycline and chlortetracycline, etc. |
| Sulfonamides | It is a synthetic antibacterial drug with a broad antibacterial spectrum and stable properties. Its main function is to affect bacterial nucleoprotein synthesis by hindering the formation of dihydrofolic acid, thereby inhibiting bacterial reproduction. This kind of drug is a derivative of p-aminobenzene sulfonamide as a skeleton structure. | Sulfadiazine, Sulfathiazole, Sulfamethoxazole, etc. |
| Quinolones | It is an artificial, synthetic antibacterial agent with a broad antibacterial spectrum and strong antibacterial activity. It uses the 4-quinolone ring as the basic skeleton structure, and acts on bacterial deoxyribonucleic acid (DNA) helicase to damage chromosomes and kill bacteria. | Enrofloxacin, Norfloxacin, Ciprofloxacin, etc. |
| β-lactams | It mainly prevents the development of cell walls. It has the greatest diversity and is capable of treating most diseases. | Penicillins and cephalosporins, etc. |
| Aminoglycosides | It mainly inhibits the production of proteins by bacteria, increases the permeability of cell membranes, and causes chemicals within the cells to flow out. It removes bacteria. | Streptomycin, gentamicin, neomycin, kanamycin, etc. |
| Macrolides | It mainly inhibits bacterial protein synthesis. | Erythromycin, Tylosin, etc. |
| Amide alcohols | Broad-spectrum antibiotics produce an irreversible bond with the 50 S component of the bacterial ribosome, which then destroys microorganisms. This inhibits the formation of DNA white matter, acyl transfer, and peptide chain extension. | Chloramphenicol, Florfenicol, Thiamphenicol, etc. |

### 4.2. ARGs in the Marine Environment

Recently, ARGs in the marine environment have mainly come from sewage treatment plants, aquaculture farms, and medical wastewater, and some also come from antibiotic production plants and animal husbandry [12,13,57]. ARGs, such as sulfonamides, tetracyclines, chloramphenicols, macrolides, and quinolones, have been found in the mariculture environment. They are usually found in mariculture and seawater sediments [59,60]. The detection abundances of sulfonamide (*sul*1 and *sul*2) and tetracycline (*tet*A and *tet*B) related to ARGs in cultured seawater are high, but there are large differences among different sampling sites [61,62]. Chen et al. [63] investigated the ARGs in 13 major mariculture farms in China and found that the absolute copy number and relative abundance of sulfonamide ARGs in cultured seawater and sediment were 4.3 and 2.3 times higher than those of tetracycline ARGs, respectively, of which the main ARG was *sul*2. To overcome the limitations of the existing ARG primers, Gao et al. [64] used Illumina high-throughput sequencing to analyze ARGs in the sediments of 10 aquaculture farms along the coast of China. The results showed that the abundance of *bac*A was the highest in all sediment samples, followed by *mex*F and *mex*B.

ARGs have also been detected in coastal waters [65–70], deep sea [71], estuary [72,73], mariculture [74–79], near shore area [80–83], polar environments [84–86], and other different marine environments. At present, resistant bacteria to tetracycline, sulfonamide, fluoroquinolone, third-generation cephalosporin, and chloramphenicol, as well as the related ARGs, have been widely detected in the marine environment [65–67,69,70]. Sulfonamide ARGs (*sul*1 and *sul*2) with high abundance were detected in the sediments of the Bohai Sea [65,66], the Yellow Sea [67], and the East China Sea [69] (Table 3), while macrolide and polypeptide ARGs were abundant in the sediments of the South China Sea [70]. Suzuki et al. [62] found that *sul*1 and *sul*2 are the main sulfa ARGs, while *sul*3 is transmitted by nonculturable bacteria in seawater. Moreover, some researchers proved that aquaculture will not only increase the abundance of ARGs in nearby waters but also increases the detection rate of multidrug-resistant bacteria in the environment far away from aquaculture areas [87–89]. Overall, ARGs are ubiquitous in the marine environment,

and with the accumulation of antibiotics and the vertical and horizontal transfer of ARGs, bacteria in the marine environment will gradually acquire resistance to antibiotics.

**Table 3.** Types and distribution of ARGs and MGEs in marine environments.

| The Marine Environment | ARGs | MGEs | Ref. |
|---|---|---|---|
| Bohai Sea | $bla_{TEM}$, *sul*1, *sul*2, *qnr*A, *tet*X, *qnr*S | *intI*1 | [65,66] |
| Yellow Sea | *sul*1, *sul*2, *tet*G, *tet*X | *intI*1 | [67,68] |
| East China Sea | *sul*1, *sul*2, *sul*3, *tet*C, *tet*W, *dfr*A1, *dfr*A13, *flo*R, $bla_{PSE-1}$ | *intI*1 | [69] |
| South China Sea | *mac*B, *acr*B, *tet*X, *bav*A, *arn*A | | [65,70] |
| Manila Bay | *sul*1, *sul*2, *sul*3 | | [71] |
| Estuary | *aac*C, *aad*A5, *mph*A, *opr*D, *opr*J, *qac*ED1, *tet*G, *aad*A1 | *intI*1 | [72,73] |
| Mariculture | *bac*A, *mex*F, *mex*B, *sul*1, *sul*2, *tet*A, *tet*B, *tet*C, *tet*D, *tet*W, *tet*Q, *tet*O, *flo*R, *qnr*A, *qnr*B, *qnr*D, *aad*A, $bla_{TEM}$ | *intI*1, *intI*2 | [62–64,74–79] |
| Beaches | *tet*A, *tet*B, *qnr*S, *sul*1, *dfr*A5, *dfr*A7 | *intI*1 | [80–83] |
| Polar | *sul*1, *sul*2, *sul*3, *qnr*B, *tet*D, *tet*G, *aad*A2, *qac*Edlta1 | *intI*1, *cInI*1, *tnp*A | [84–86] |

## 5. ARGs in the Ballast Water

### 5.1. ARGs in the Ballast Water of Ships

In recent years, the proliferation of ARGs has gradually evolved into a serious global problem [5], and ARGs have been detected in sewage, rivers, inshore fishing grounds, marine sediments, and inshore seawater [56,60–64,90]. Seawater containing ARGs and human bacterial pathogens (HBPs) will be pumped into the ballast tank of the ship and then discharged to the port sea area of the destination [91]. This new pollutant can be transmitted to all parts of the world through ship ballast water, and the spread of ARGs will enhance the resistance of pathogenic bacteria to antibiotics.

Lv et al. [92] found that ARBs were detected in 25 out of 30 ship ballast water samples from different watersheds, with a detection rate of 83.3%; among these, penicillin-resistant bacteria had the highest detection rate, but their absolute abundance was relatively lower than that of hygromycin, chloramphenicol, tetracycline-like and vancomycin-resistant bacteria (Figure 1). According to the articles that were published, it seems that antibiotic stress may not be a particular technique to increase ARG production [92].

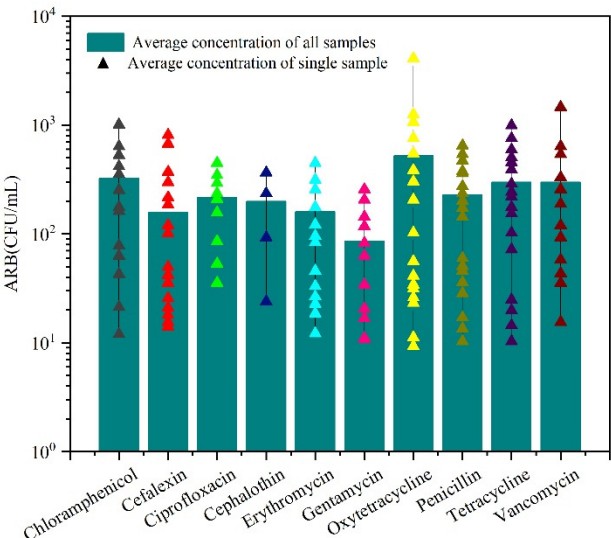

**Figure 1.** Ten antibiotic-resistant bacteria (ARB) were found in ballast water [92].

Penicillin was the first antibiotic discovered by human beings in the world, and it has been used for a long time. Therefore, the amount of penicillin-resistant bacteria in the global marine environment is relatively high [93]. However, other inexpensive and effective antibiotics, such as oxytetracycline, vancomycin, and tetracycline, have gradually

developed and are now commonly used in aquaculture [94,95]. These antibiotics have a long half-life, slow metabolism, and nondegradable properties, which make them remain in the environment and continuously enrich ARGs. These persistent ARGs will aggravate the risk of ARG transmission in the environment after being discharged into the adjacent sea area. Schwartz et al. [96] concluded that the prevalence of vancomycin- and tetracycline-resistant bacteria is higher than that of penicillin-resistant bacteria.

The abundance of different ARGs in ballast water varies greatly. Lv et al. [92] found that the abundance of gentamicin-resistant bacteria was relatively lower than that of other ARBs in ballast water. Mohamed-Hatha et al. [97] and Souissi et al. [98] also proved that the abundance of gentamycin-resistant bacteria was low in marine environments. However, Lv et al. [99] conducted research on the ballast water of 28 ships in Chinese ports and proved that *sul*1 was the most abundant ARG with a concentration of $5.0 \times 10^8$ copies/mL. Niu et al. [66] and Stoll et al. [100] also found that *sul*1 had a high absolute abundance in surface water, and sulfonamide resistance genes dominated and persisted. Due to their widespread use, high discharge rate, high solubility, and persistence of sulfonamide antibiotics, sulfonamide-resistant bacteria can be gradually enriched in ballast water and persist in the aqueous environment for 5–10 years in the absence of selective pressure [73]. Quinolone antibiotics are a newer type of antibiotic, but bacteria that are resistant to them have been found in different types of seawater and sediments, which could be dangerous [101].

Many studies found that the ARG content in ballast water was higher than that in the samples collected in other environments [99,102]. Ballast water samples contained higher concentrations of ARGs than those in the Yellow Sea and Bohai Sea [61,103]. William et al. [47] found higher concentrations of *tet*M in ballast water than in seawater, suggesting the potential for ballast water to promote the global diffusion vector of ARGs. Furthermore, many HBPs have been detected in estuaries [73,104], ports [65,66], and coastal waters [59–63], which contain ARGs. The discharge of ballast water containing ARGs and HBPs into the seaport sea area of the destination will cause certain pollution risks to the water environment of the seaport area. The spread of ARGs will enhance the resistance of pathogenic bacteria to antibiotics, and HBPs will make humans sick [105,106]. Therefore, the prevention and control of pathogenic bacteria in ballast water have ushered in new challenges. The long-term existence and spread of ARGs in ballast water lead to the growth and spread of the drug resistance of pathogenic bacteria.

*5.2. ARGs in Ballast Water Sediments*

Any material that settles out of the ballast water tends to be deposited in the sediment at the bottom of the ballast tanks. Bottom sediments may contain a variety of marine organisms and microorganisms in their active and dormant stages, including pathogens. Prange et al. [107] found that the amount of sediment in the ballast tank of a merchant ship measured up to 200 tons. Living and nonliving organisms in ballast sediments need to be managed by appropriate methods; otherwise, they will pose risks. Hamer et al. [108] indicated that the thickness of the sediments in the double-bottomed ballast tanks of cargo ships increased to 30 cm after only two years of use. Most of the water left in the ships and sediments may contain non-native organisms. Gollasch et al. [109] recorded 990 species in ballast water sediment samples. Drake et al. [33] separated particle sludge that contained a variety of organisms in the ballast water sediment. Therefore, the ballast tank sediment provides a solid matrix rich in nutrients. The sediment in the ballast water tank will cause serious damage to the ecological environment of docks and port waters if it is not reasonably and effectively disposed of. More seriously, microorganisms that reside in sediments may exchange ARGs with one another. This has the potential to seriously affect the area surrounding the shipyard.

Lv et al. [28] studied ballast tank sediments and detected 10 ARGs in the sediments, among which the sulfonamide resistance genes (*sul*1 and *sul*2) had the highest absolute abundance, ranging from $8.36 \times 10^7$–$1.61 \times 10^9$ copies/g to $2.55 \times 10^7$–$4.31 \times 10^8$ copies/g,

respectively (Figure 2). Sulfonamide is the most commonly used antibiotic in marine aquaculture. Due to the particularities of aquaculture, the utilization rate of antibiotics is low, and the high concentration of antibiotics is eventually released into the surrounding marine water body or the sediments. Sulfonamide continues to spread and migrate, which, in turn, pollutes the marine environment [33] and affects coastal areas and marine sediments [110]. Ballast operations generally occur in offshore waters. Because of the relatively high abundance of *sul*1 and *sul*2 genes in offshore waters, *sul*1 and *sul*2 can be enriched in sediments. Lv et al. [28] found that the abundance of sulfonamide resistance genes was significantly higher than that of other ARGs, such as tetracycline resistance genes (*tet*M and *tet*Q). Chen et al. [69] and Lin et al. [111] found that tetracycline resistance genes were more easily removed than sulfonamide resistance genes. Gao et al. [112] investigated the pollution of ARGs in the aquaculture environments of Tianjin and proved that the abundance of sulfonamide resistance genes was much higher than others. There is a strong link between antibiotics and the ARGs they are linked to within the ballast water sediment environment. This means that antibiotic residues and ARGs may end up in the marine environment through the circulation of seawater, the suspension of sediment, and the movement of organisms. Because of this, it is important to constantly monitor the distribution and abundance of ARGs in ballast water, as well as develop appropriate methods to hinder the spread of ARGs via ballast water.

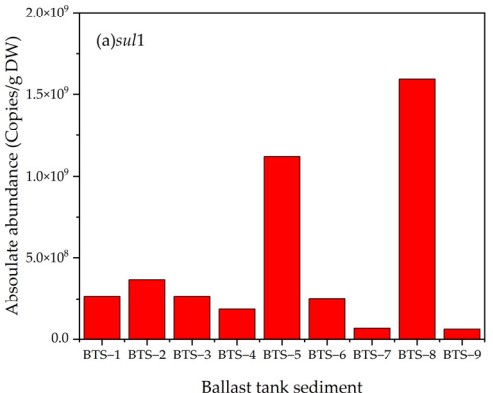 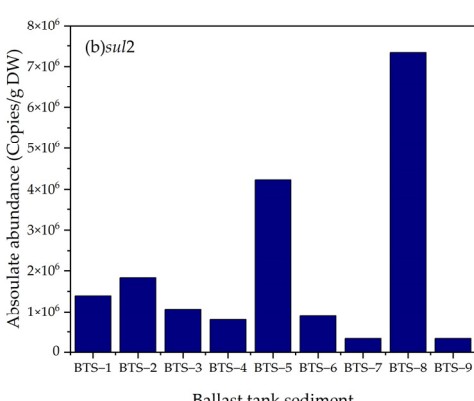

**Figure 2.** Absolute concentrations of (**a**) *sul*1 and (**b**) *sul*2 in ballast tank sediments (BTS−1 to BTS−9 refer to ballast tank sediment samples) [28].

### 5.3. Influencing Factors of Resistance Genes

Recently, many studies have focused on the influence of a single type of antibiotic or environmental pressure on the distribution and spread of ARGs, aiming to reduce the ARGs and provide methods for the management of ARG contamination [33,109]. However, a variety of parameters, including physical factors and chemical factors, and specific components like temperature, pH, and alkalinity, all have an impact on the ARGs distributed in aquatic environments.

Some traditional pollutants, such as heavy metals, are selected to enrich antibiotic resistance coselection. It has been shown that the abundance of ARGs in the environment is related to the concentration of heavy metals [28]. Heavy metals can directly act on the microbial cell structure or indirectly affect bacteria by changing environmental conditions (such as pH, etc.), resulting in toxic effects on the bacteria. The long-term selective process of heavy metals can stimulate the self-defense mechanism of bacteria and gradually develop resistance. He et al. [113] found that both heavy metals and antibiotics can promote the transfer of resistance genes, and their combined effect is greater than their individual effects. Manaia et al. [25] believe that the influence of antibacterial residues or metals is the determinant of the fate of ARBs and ARGs in wastewater treatment. Yang et al. [20] summarized the distribution of antibiotics and ARGs in global lakes. It was found that chemical pollution, including antibiotics and heavy metals, affected the distribution of

ARGs in lakes through selective pressure. Bacterial community is the main factor in forming drug resistance, followed by mobile genetic elements. Therefore, there was a certain correlation between heavy metal content and ARGs that influenced the abundance of ARGs.

Integrons (such as *int*I) often co-occur with antibiotic resistance genes due to the insertion of the antibiotic resistance gene. Lv et al. [28] found that *int*I was much more abundant than ARGs in shipyard sediments. The correlation between intI and ARGs was very strong. The high viability of *int*I enhances the propagation of ARGs through the horizontal gene transfer between microorganisms. Additionally, Zhang et al. [114] found that subinhibitory disinfectant doses may enhance the transfer of genes that cause bacteria to become resistant to antibiotics from one type of bacteria to another.

The content and distribution of antibiotic resistance genes in coastal waters are affected by human activities and environmental factors. Windil et al. [26] found that aquaculture changed the characteristics of ARGs and mobile genetic elements in Baltic Sea sediments. Lupo et al. [115] found that environmental pollution and water quality parameters affected the antibiotic resistance of bacteria in surface water. Rizzo et al. [23] and Bouki et al. [24] summarized the fate of ARB and ARGs in urban sewage treatment plants and believed that the effluent from urban sewage treatment plants was the main source of antibiotic release into the environment, causing ARGs and ARB to diffuse into the water environment. Antibiotic residues are the main driving force for the maintenance and transmission of ARB and its genes in wastewater. Moreover, it is also possible to change the formation and transmission rates under the action of antibiotics or heavy metals [59,116,117]. It is important to consider how the combined impacts of various antibiotics or combinations of antibiotics and environmental factors affect the characteristics of resistance gene contamination.

## 6. Changes in ARGs in Ballast Water after Disinfection

### 6.1. Ballast Water Disinfection

Now, ARGs have been found in a variety of water environments [118,119], but there is no special monitoring or control method for antibiotics and resistance genes. Many studies have focused on the removal of antibiotics and resistance genes in sewage treatment plants [120,121], while there are few studies on the treatment of antibiotics and resistance genes in ballast water. Therefore, it is necessary to emphasize the control of ARGs in ballast water. This chapter summarizes the research status of ARG-removal technology in aquatic environments in recent years and compares the advantages and disadvantages of each technology. Table 4 summarizes the removal effects of several treatment methods on ARGs and compares their advantages and disadvantages in detail.

**Table 4.** Advantages and disadvantages of different ballast water disinfection methods and removal effect of ARGs.

| Classification | Advantage | Shortcoming | Removal Effect of ARGs | Ref. |
|---|---|---|---|---|
| Ultraviolet disinfection | No byproducts, no chemical residue, noncorrosive water treatment. | There is photoreactivation. It is difficult to achieve a higher UV dose in actual production. | At a UV dose of 5 mJ cm$^{-2}$, erythromycin ARGs and tetracycline ARGs decreased by an average of $3.0 \pm 0.1$ log and $1.9 \pm 0.1$ log, respectively. UV dose of 200~400 mJ cm$^{-2}$ reduced ARGs (*mec*A, *van*A, *tet*A and *amp*C) by 3~4 log; weak effect on *tet*X, *sul*1, *tet*G, and *int*I1. At 60 min, the removal efficiency of *sul*1-qPCR is greater than 3.50 log, and that of *int*I1-qPCR is greater than 4.00 log. | [41,122–124] |
| Chlorination disinfection | Simple operation, low cost. | It produces toxic byproducts, causes chemical residues. | Reduce erythromycin ARGs of $0.42 \pm 0.12$ log and tetracycline ARGs of $0.10 \pm 0.02$ log. *sul*1, *tet*X, *tet*G, and *int*I1 can remove 1.30~1.49 log. When the chlorine dosage is from 5–20 mg L$^{-1}$, the removal of ARGs increases slowly, and a larger free chlorine dosage will lead to higher gene removal efficiency. | [42,122,123] |
| Ozonation | Strong oxidizability, fast reaction speed, no residue. | Ozone is easy to consume. It produces harmful byproducts. | *sul*1 and *tet*G decreased by 1.65~2.28 log. Under high ozone concentration, the removal rate of *bla*$_{ctx}$ was slightly higher than that of *qnr*S gene. | [125,126] |

**Table 4.** *Cont.*

| Classification | Advantage | Shortcoming | Removal Effect of ARGs | Ref. |
|---|---|---|---|---|
| Fenton oxidation | Strong oxidation, simple operation, and low cost. | Reagent consumption is high. Excessive $Fe^{2+}$ will affect the effluent quality and cause secondary pollution. | With an increase in Fenton reagent concentration, the band strength of tetM gene decreased gradually. The gel electrophoresis band strength of *tet*M decreased with an increase in oxidant dose; under the action of $Fe^{2+}/H_2O_2$, the concentrations of *sul*1, *tet*X, and *tet*G decreased by 2.58~3.79 log, while under the action of $UV/H_2O_2$, they decreased by 2.8~3.5 log. | [43,127,128] |
| Photocatalytic oxidation | Tasteless and nontoxic, strong oxidation ability and complete sterilization, mild reaction conditions. | With the progress of the reaction, the catalyst will be deactivated, and the utilization rate of the light source will not be high. | 5.8 log *mec*A and 4.7 log *amp*C were removed. The degradation efficiency of *flo*R, *tet*C, *sul*1, and *int*I1 was 97.82, 20.66, 99.45, and 93.67% respectively. The synthesized graphene-based photocatalyst successfully removed *amp*C and significantly reduced the *ecf*X abundance of *Pseudomonas aeruginosa*. | [44,45,67,129,130] |

## 6.2. Changes in ARGs in Ballast Water under Different Disinfection Technology

### 6.2.1. Chlorination Disinfection

Chlorination disinfection usually uses free chlorine and chlorine dioxide as disinfectants to eliminate pollutants, which is an important disinfection method for water treatment. Studies have shown that when the free chlorine concentration is 5 mg/L, 10 mg/L, and 15 mg/L, the pollutants, such as penicillin and tetracycline, in the urban sewage environment can be completely removed [131]. Chlorination may produce highly toxic disinfection byproducts. After chlorination disinfection, the toxicity of quinolones to photobacteria is enhanced [121]. In addition, a large amount of toxic disinfection byproducts can be generated after the chlorination disinfection of tetracyclines [132,133].

Free chlorine solution can act on target components such as DNA, RNA, and enzymes in the cytoplasm, causing the cleavage of resistance gene fragments [134]. Disinfection with higher doses of chlorine not only inactivates most donor bacteria but also damages the structure and function of the donor bacteria. Surviving donor bacteria have reduced extracellular secretions during the splicing process, making it difficult to transfer plasmids into the recipient bacteria, resulting in a significant reduction in the frequency of splice transfer [135]. However, it has also been proven that there was no significant removal of tetracycline resistance genes and erythromycin resistance genes after free chlorine disinfection (15 mg $Cl_2$ min/L) [136]. In addition, some research indicated that a high concentration of free chlorine (300 mg $Cl_2$ min/L) does not affect a decrease in resistance genes [42]. Moreover, some scholars claim that chlorination disinfection may promote the enrichment of resistance genes [137]. Liu et al. [138] found that the concentration of a-eARG increased 7.8 times after chlorine disinfection. It shows that the ARG released into the extracellular parts after chlorine disinfection tends to combine with the substances in the water to form a-eARG.

### 6.2.2. Ultraviolet Disinfection

Ultraviolet disinfection technology is widely used in water treatment because of its good effect, convenient use, and lack of chemical additions. Some studies indicate that ultraviolet (UV) irradiation can effectively remove antibiotics from aqueous solutions, especially antibiotics that are difficult to degrade [139]. UV can efficiently degrade lactams, fluoroquinolones, and tetracyclines [140].

Guo et al. [41] showed that, after UV treatment at an energy density of 5 mJ/cm$^2$, the proportion of erythromycin-resistant bacteria in wastewater decreased, while the proportion of tetracycline-resistant bacteria increased. This result indicates that tetracyclin-resistant bacteria show higher tolerance to ultraviolet radiation and extrapolates that UV has a selectivity effect on ARBs [141,142]. In subsequent research, Guo et al. [143] found that UV disinfection (>10 mJ/cm$^2$) can greatly inhibit horizontal gene transfer (HGT) and directly damage plasmid-containing ARGs, resulting in the death of the HGT donors and receptors. The above research shows that low-dose UV disinfection can effectively

reduce ARGs in a water environment, but other researchers have reached the opposite conclusion. McKinney et al. [123] believed that high-dose UV disinfection could effectively remove ARGs. Through a series of experiments, they concluded that the UV dose required to reduce 3~4 log ARGs (*mec*A, *van*A, *tet*A, and *amp*C) was 200~400 mJ/cm$^2$ (Figure 3). However, the UV radiation dose in municipal sewage treatment plants is usually lower than 10 mJ/cm$^2$ [143]. Therefore, in actual water treatment equipment, it is very difficult to achieve such a high UV dose. Also, ampicillin-resistant genes (*amp*C) are very resistant to UV disinfection, so this method cannot be applied to all ARGs [123].

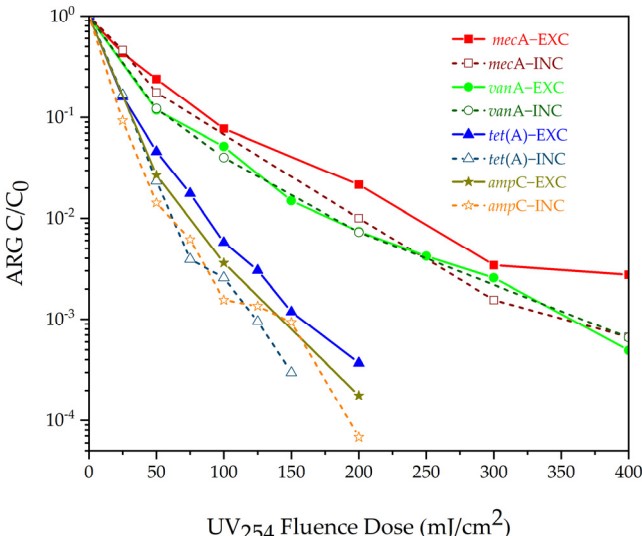

**Figure 3.** UV (λ = 254 nm) disinfection curves for antibiotic resistance genes (ARGs). INC = Intracellular. EXC = extracellular [123].

6.2.3. Ozone Disinfection

Ozone is a strong oxidant that can react rapidly with organics in the form of direct oxidation. It can also indirectly oxidize most organic compounds with the hydroxyl radicals (OH) generated by water substrates [144]. In solutions of amoxicillin, doxycycline, ciprofloxacin, and sulfadiazine, the removal rate of the antibiotics accelerated with a continuous increase in the ozone concentration. Ozone (75 mg/L) can remove 95% of antibiotics, and its decomposition byproducts have no antibacterial activity or toxicity [145]. A total of eight antibiotics were completely removed when the ozone dose reached 125 mg $O_3$/g for dissolved organic carbon (DOC) and the hydraulic retention was 40 min [146]. When the amount of ozone reached 657 mg/L and the water retention time was 120 min, 96% of the oxytetracycline was removed [146].

Compared with UV disinfection, ozone has a higher removal efficiency for antibiotics, yet it is easy to produce ozone byproducts that accumulate in water bodies. Ozone and ·OH oxidize and destroy bacterial cells. They then destroy the bacteria's organelles, DNA, RNA, and other parts, directly [147]. Ozone disinfection kills a wide range of resistant bacteria, which causes the abundance of *Escherichia coli*, *Staphylococcus*, and *Enterococcus* to decrease sharply (0.73 mg $O_3$/mg DOC, 20 min) [148]. Also, ozone treatment with 0.9 g $O_3$/g DOC killed 60.2–98.9% of resistant bacteria like *Enterococcus*, *Staphylococcus*, *Enterobacter*, and *Pseudomonas aeruginosa*, with the abundance of ARGs, such as *bla*VIM, *van*A, *amp*C, and *erm*B, decreasing by 18.7, 49.9, 69.8, and 99.3%, respectively [149]. Compared to UV disinfection and chlorination disinfection, ozone disinfection is better for eliminating resistant bacteria and resistant genes [123]. This may be because when a cell membrane is damaged, and proteins leak out, microbial gene fragments are exposed directly to ozone.

### 6.2.4. Free Radical Disinfection

Free radical disinfection techniques include photocatalysis and the Fenton reaction. Photocatalytic oxidation has a high removal rate of antibiotics and resistance genes in water and is a widely used method in free radical disinfection technology. Combining photocatalysis disinfection with the simultaneous dosing of UV irradiation (18.9 mJ/cm$^2$) and H$_2$O$_2$ (40 mg/L) can completely degrade antibiotics such as ampicillin, erythromycin, and tetracycline [150]. Research on the elimination of antibiotics and their transformation products in wastewater suggested that the antibiotics in wastewater could be completely removed with a UV irradiation dose of 0.9 kJ/L for 90 min. Moreover, when UV was used in combination with H$_2$O$_2$/Fe, antibiotics could be completely removed within 30 min. In addition, research also shows that adding low doses of persulfate can get rid of antibiotics completely in 7–18 s [151].

Photocatalytic disinfection also effectively degrades resistance genes. Under UV/H$_2$O$_2$ treatment for 30 min, the abundance of resistance genes was reduced by 2.8–3.5 log [152]. Over the same treatment time, UV disinfection and chlorine disinfection could only reduce the abundance of resistance genes by 0.80–1.21 log and 1.65–2.28 log, respectively [122]. Although these results showed that UV/H$_2$O$_2$ treatment had a high removal rate for resistant bacteria and ARGs, the removal effect of free resistance genes in suspension was not obvious [153,154]. This could be due to a low oxidant dose or the inability of OH to react with other intracellular substances on DNA fragments [155,156]. When comparing OH to other disinfection methods, free radical technology is effective for antibiotic removal, resistant bacteria inactivation, and resistance gene degradation. However, because of the requirement for additional catalysts, free radical disinfection costs more than other methods, like UV and chlorination, because it needs more catalysts.

A single disinfection method cannot completely mineralize antibiotics or inactivate ARGs, and the continuous formation of disinfection byproducts may produce serious environmental toxicity. Some methods of disinfection, such as UV/H$_2$O$_2$ and UV Fenton, may be better for the removal of antibiotics and genes that make bacteria resistant to them. Zhang et al. [128] found that under optimal conditions, the UV/H$_2$O$_2$ method could reduce the abundance of resistance genes by 2.63–3.48 orders of magnitude. Hou et al. [157] found that among the treatment technologies, such as UV, ozone oxidation, Fenton oxidation, and Fenton/UV, Fenton oxidation and Fenton/UV are the most effective methods to remove the total bacteria and ARGs in water. Giannakis et al. [43] conducted a study on the elimination of antibiotic-resistant bacteria and ARGs by sunlight and the sunlight/Fenton method and found that both can effectively remove ARGs, and the removal rate of the sunlight/Fenton method was faster than that of the sunlight/Fenton method. Also, more attention should be paid to the byproducts of disinfection that are generated during the process. The ozone disinfection method can effectively remove antibiotics and resistance genes, but it can also easily produce disinfection byproducts. The UV photolysis of antibiotics may also generate more toxic intermediate products, thereby causing toxic effects on other organisms in the aquatic environment [158].

### 6.3. The Affect of Subinhibitory Concentrations of Disinfectants

The long-distance distribution of residual disinfectants through pipelines often decreases to subinhibitory levels (below minimum inhibitory concentrations [MICs]) or even becomes undetectable. Residual disinfectant in the retransmission pipeline is used to inhibit the regeneration of microorganisms in the pipeline. Therefore, ARGs and ARGs in treated ballast water should be exposed to relatively low concentrations of disinfectants. After a long voyage, the residual disinfectants in the ballast water may be at subinhibitory levels. The concentration of the disinfectant may also promote the transfer of antibiotic resistance. Xi et al. [159] found that ARGs were more abundant in tap and reclaimed water than in freshly-treated water. Anderssen et al. [160] indicated that subinhibitory levels of antibiotics could induce the transfer of ARGs, which may have an important impact on the spread of ARGs by participating in the reactive oxygen species (ROS) response system and

SOS response pathway [114]. Guo et al. [143] investigated the effects of UV disinfection and chlorination on the transfer of ARGs and proved that UV disinfection and chlorination had a significant effect on their conjugate transfer (Figure 4). In contrast, the frequency of conjugation significantly increased with the subinhibitory dose (of up to 40 mg $Cl_2$ min/L). Ye et al. [114] found that subinhibitory concentrations of disinfectants promote the conjugation transfer of ARGs within and between genera. Subinhibitory concentrations of chlorine, chloramine, and $H_2O_2$ promoted intragenus conjugation transfer by 3.4–6.4, 1.9–7.5, and 1.4–5.4 times, respectively (Figure 5) [114].

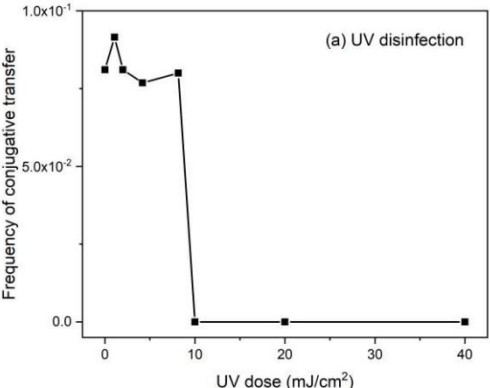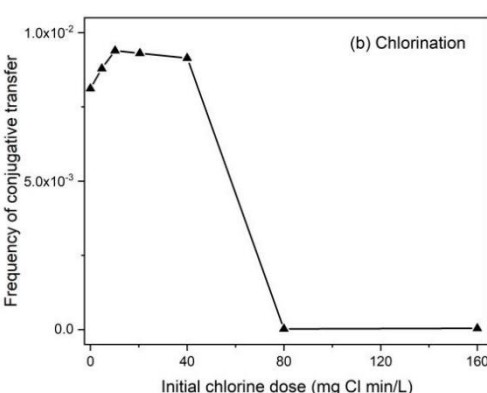

**Figure 4.** Effect of (**a**) UV disinfection and (**b**) chlorination on the frequency of binding transfer [143].

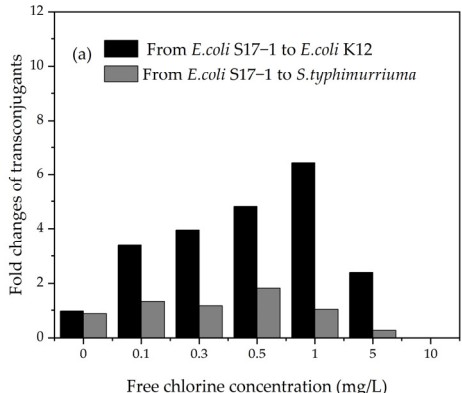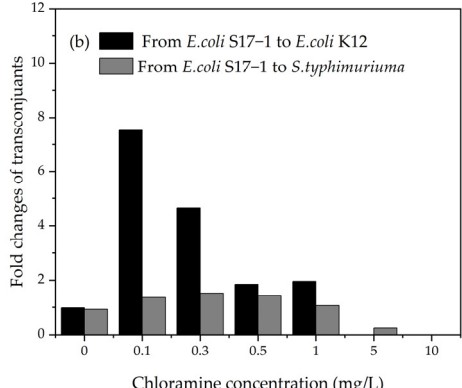

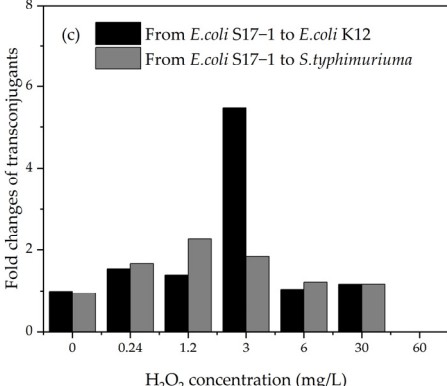

**Figure 5.** Effect of (**a**) free chlorine, (**b**) chloramine and (**c**) $H_2O_2$ on intrageneric and transgeneric conjugation transfer [114].

## 7. Conclusions

This paper summarizes the main types of ARGs in ballast water and ballast tank sediments and analyzes the factors that influence resistance genes. In addition, it analyzes the changes in ARGs after disinfection. The conclusions are shown as follows.

1. Several resistance genes have been discovered in ballast water and sediments. This is generally due to the widespread use of antibiotics in aquaculture;
2. The levels of ARGs in ballast water were higher than those in the samples collected from nearby marine environments. ARGs were found in ballast water samples from different sea areas. This result indicated that ballast water could promote the spread of ARGs, which should be further considered in the formulation of ballast water discharge standards;
3. Disinfection treatment can enhance the removal of ARGs in ballast water. However, if the disinfectants are reduced to subinhibitory levels, a potential mechanism for the conjugate transfer of the ARGs will emerge.

ARG pollution is widespread in the water environment, and its horizontal gene transfer mode in the water environment is a hot topic of current research. However, there is little research on the transfer of ARGs in ballast water environments. Future research needs to further investigate the impact of ballast water discharge on the transmission of ARGs in local waters. Furthermore, some studies have shown that disinfectants with subinhibitory concentrations during water transportation can promote the conjugation and transfer of ARGs within and between genera. Therefore, it is necessary to determine the subinhibitory concentration of a disinfectant that causes the horizontal transfer of ARGs in ballast water. At present, most of the studies on ARG removal regarding disinfection technology report a single technology; there are few studies on ARG removal using more than two combined processes. The removal mechanism of ARGs by each disinfection process is different, and combination processes may produce synergistic effects.

**Author Contributions:** Writing—original draft preparation, J.G.; writing—review and editing, J.G. and S.; visualization, J.G. and Y.Z.; editing, B.J., C.W. and Y.Z.; supervision, Y.Z. and C.L. All authors have read and agreed to the published version of the manuscript.

**Funding:** This work was founded by Natural Science Foundation of Shanghai, grant number No. 20ZR1438200, and the National Science Foundation of China, grant number No. 51778565.

**Institutional Review Board Statement:** Not applicable.

**Informed Consent Statement:** Not applicable.

**Data Availability Statement:** Not applicable.

**Conflicts of Interest:** The authors declare no conflict of interest.

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
