# Peer review of "Review of the Distribution and Influence of Antibiotic Resistance Genes in Ballast Water"

_water, doi:10.3390/w14213501_

Round 1
Reviewer 1 Report (Previous Reviewer 3)
The revised manuscript can be accepted after the following points are updated.
1. The brackets in x-axis of Figure 1 is not consistent with the font.
2. Figure 3 is not mentioned properly in the text. Also, the units of x & y-axis of Figure 3 are missing, please update.
Author Response
Revision Notes for " Review of the distribution and influence of antibiotic resistance genes in ballast water (Manuscript ID: water- 1925376)"
Responses to Reviewer #1’s comments:
The revised manuscript can be accepted after the following points are updated.
- The brackets in x-axis of Figure 1 is not consistent with the font.
REPLY: Many thanks for the reviewer's comment. Picture 1 has been redrawn, and it has been modified according to your comments.
MODIFICATION:
- Figure 3 is not mentioned properly in the text. Also, the units of x & y-axis of Figure 3 are missing, please update.
REPLY: Many thanks for the reviewer's comment. Picture 3 has been redrawn, and it has been modified according to your comments. The reference to Figure 3 is supplemented in the text.
MODIFICATION:
McKinney et al. [103] believed that high-dose UV disinfection could effectively remove ARGs. Through a series of experiments, they concluded that the UV dose required to reduce 3~4 log ARGs (mecA, vanA, tetA and ampC) was 200~400 mJ/cm2 (Figure 3).

Reviewer 2 Report (New Reviewer)
The manuscript entitled “Review of the distribution and influence of antibiotic resistance genes in ballast water” contains 139 references and represents a comprehensible review of the literature that is currently lacking in this area of research.
1) A large part of the manuscript is focused on the influence of disinfection and several other physicochemical factors on the transport/formation of antibiotic resistance. This should also be reflected in the graphic abstract to get more attention.
2) In a review, a large part of the conclusion should focus on future perspectives and recommendations that future readers will greatly appreciate.
3) I would suggest expanding the section of the introduction dealing with sources of antibiotic resistance genes in ballast water. An additional section focusing on the fate of antibiotics is optional, but it will increase the attention of the current manuscript.
4) Reference style needs to be corrected. Journal chapter and page numbering is missing in several references.
5) Some of possible references for ARGs and connected environmental fate:
10.1016/j.scitotenv.2013.01.032
10.1016/j.ecoenv.2013.01.016
10.1016/j.envint.2018.04.011
10.1016/j.envint.2018.03.044
Author Response
Please see the attachment.

This manuscript is a resubmission of an earlier submission. The following is a list of the peer review reports and author responses from that submission.
Round 1
Reviewer 1 Report
General Comments:
This manuscript reviews distribution and influence of antibiotic resistance genes in ballast water environment. The topic is important for better management of ballast water since tons of ballast water is discharged annually into the marine environment. However, the manuscript is written incompletely and not easy to be understood. Grammatical errors found throughout the manuscript. Figures and Tables are inappropriately presented. Abbreviations are not defined when they are used for the first time. Besides, the manuscript is not written according to ‘Instructions for Authors’ given by the journal. Detailed comments are in ‘Specific comments’ below.
Specific Comments:
. Abstract: Please check the last sentence. I think that the authors meant ‘the transfer of ARGs’ instead of ‘the removal of ARGs.’ The abstract should be a total of about 200 words maximum according to the journal instruction for authors. However, the abstract of manuscript has more than 210 words.
. Reference format in the manuscript does not follow ‘Instructions for Authors’ given by the journal. Reference numbers in the text should be placed in square brackets [ ], and placed before the punctuation; for example [1] ], [1–3] or [1,3] according to the instructions. However, reference numbers in the manuscript are in superscript and [ ]-[ ] are found instead of [ - ]. Reference list is not in the style recommended by the journal instructions as well.
. Gene names should be italicized according to general rules of gene nomenclature. However, many gene name in the manuscript are not italicized.
. Inappropriate presentation of bacterial names: Bacterial names should be italicized according to nomenclature of bacteria. E.g., Escherichia coli (E. coli), Vibrio cholera etc.
. Table 1: Define * in the table. Check and revise the header of the table. The International System of Units (SI) prescribes inserting a space between a number and a unit of measurement. One space has to be left between a word and ( ). What does ‘IMOD-2 Standard’ in the title of Fig.1 mean? IMO D-2 Standard?
. It is important that figures in the manuscript should have shown their sources. Define abbreviations in X axes of Fig. 1-3. Fig. 3 needs to define graph legends such as RW, UV5, and UV10. X axis of Fig.4 needs to correct ‘cm2.’ ‘Binding transfer’ in Fig. 4 title should be ‘conjugation (or conjugative) transfer.’ ‘Conjugation’ is right term although ‘binding’ has similar meaning.
. 2.2.3. Other biological: Incomplete title. The paragraph under the title needs reference(s).
. Abbreviations: The manuscript does not define abbreviations when they appear for the first time. Abbreviations should be defined for the first time they appear in each of three sections: the abstract; the main text; the first figure or table. When defined for the first time, the abbreviation should be added in parentheses after the written-out form. Once abbreviations are defined, they can be used later on.
. Grammatical errors are found throughout the manuscript. For example, page 9, lines 323-326.
Reviewer 2 Report
Based on the study of resistance genes in ballast water, this paper discusses the types and abundance of resistance genes in ballast water and sediment, and expounds the influence of disinfection on resistance genes in ballast water. The research process is detailed, the logical thinking is clear, and the conclusion is basically feasible. I suggest the paper could be minor revised. The detailed comments are as following:
1. The format of some references is incorrect, and it is suggested to modify them.
2. There are some format problems in the article. For example, the indentation of the second paragraph of the introduction is inconsistent with that of other paragraphs. At the end of the second paragraph, a period is added at reference [6]. It is suggested to modify the format.
3. Unify the title format. The title format of Table 1 is inconsistent with that of Table 2 and Table 3. It is suggested to refer to the format required by the journal for modification.
4. There is a clerical error in the text. For example, in sections 3.1 and 5.2, there is a writing error "ARBs", which is correctly written as "ARGs". It is suggested to modify. In Section 3.2, "see Table 2" is suggested to be modified to "Table 2".
5. Explanation of repeated abbreviations in the text. The abbreviations of antibiotic resistance genes (ARGs) have been introduced in the abstract, but the introduction of the abbreviations of ARGs is repeated in Section 3.1. It is suggested to modify it.
6. There is an error in the corner mark of references. In the last paragraph of part 4.1, there is an error in the corner mark of references [56][60] and [61] in the first sentence, which is suggested to be modified.
7. Text format error. According to the format requirements, antibiotic resistance genes should adopt the inclined format. For example, "tetM , sul1 , tetQ" appears in the text. In this paper, some of them adopt inclined format, and some of them do not. It is suggested that the format should be unified and modified according to the format required by the journal.
8. The use of spaces is inconsistent. There are some inconsistencies in the use of spaces in the article. For example, in the citation of references. It is suggested to revise according to the requirements of the journal.
Reviewer 3 Report
Dear Editor,
Thanks for inviting me for review the manuscript entitled Distribution and influence of antibiotic resistance genes in ballast authored by Jiaqi Guo et al.
The manuscript investigated the situation of ballast water discharge and its negative effect of biological invasion to marine ecosystem, analysed the ARGs distribution and influencing factors in ballast water environment, and summarized the changes of ARGs before and after disinfection by some common techniques. The content of this manuscript is informative, the manuscript could be accepted after the following issues are addressed.
1) The relevant references should be cited following the captions of each figure.
2) The abbreviation of for antibiotics appeared in Figure 1 should be mentioned in the text to avoid confusing for readers.
3) As-mentioned in page 6, the author stated “Penicillin-resistant bacteria had the 175 highest detection rate, but their absolute abundance was relatively lower than that of hygromycin, chloramphenicol, tetracycline-like and vancomycin-resistant bacteria (Figure 1)”, according to Figure 1, only CHL, TET, and VAN have higher ARB values compared to PEN, please check the value for hygromycin, and update.
In addition, the “ARB” on y-axis of Figure 1 is not mentioned in the text, the caption of Figure 1 should be rewritten as “The ARB of Ten antibiotic-resistant bacteria in ballast water”.
4) It is suggested to have a drawing on ARGs distribution in ballast water and sediments according to existing articles so that readers could have a clear understanding, this will also make the article be more attractive.
5) Minor revisions
· Font of Figures should be checked and revised to keep consistent.
· The label (a) and (b) in Figure 2 should be placed at the up left or right left, instead of at the bottom.
· Minor ticks in Figure 2 should be removed as it does not give any meaning for the Colume.
· “RW” in Figure 3 did not mention in the text, it will confuse readers, please check.
· The maximum value of x-axis in Figure 5 is given to 60, please shorten it so that it won’t leave too much blank at the end.
Reviewer 4 Report
Comments for water-1833783: Reject
Dear editor,
I have reviewed the manuscript titled “Distribution and influence of antibiotic resistance genes in ballast water environment”. Ballast water can be considered as a carrier for the migration of antibiotic resistance genes (ARGs) at long distance, since it promote the exchange of water directly between geographically isolated sea areas. This paper summarize the occurrence and abundance of ARGs as well as antibiotic resistant bacteria in the ballast water and sediment based on several reports, but it is not a qualified review.
Generally, the author should correct all spelling, grammar and spacing mistakes. Some sentences seem to have been translated from Chinese references. Moreover, the structure of the paper is chaotic, and many irrelevant contents are pieced together. For example, the section of “5. Changes of ARGs before and after disinfection”, has nothing to do with the ballast water. Besides, the authors should also give the reasonable reference for all the Figures. In my opinion, this article does not contribute to the field of ballast water and ARGs.
The detailed comments are in the following.
1. Line 32-39, please give the reference for those sentence.
2. Line 55-58, confused, please rewrite that sentence.
3. Line 58-61, the name of the pathogenic species should be italicized, please check that.
4. Line 60, negative? It seems to be positive.
5. Line 62-65, give the reference and rewrite that.
6. Line 67-69, it is repetitive with the Line 132-134.
7. Line 77, “2. Ballast water”, in deed, this section was irrelevant with the topic of ARGs.
8. Line 89-90, the writing is not fluent, rewrite that sentence.
9. Line 114-117, the conclusion is somewhat arbitrary.
10. Line 119-129, please give the reference. The author should quote appropriately and give reasonable sources.
11. Line 134-137, 150-152, 157-160, rewrite those sentences.
12. Line 147-148, I’m not sure the conclusion is correct. Antibiotics and ARGs originated from land-based contamination was also the main driver of ARGs.
13. Line 163-166, repeat with the Line 42-46, and please give the reference.
14. Line 190-193, please give the reference, and the connection between those contexts is not obvious.
15. Line 212-216, give the reference.
16. Line 224-225, 289-292, rewrite that sentence.
17. Line 293-294, really? Many studies have reported the influence of ARGs in various environment.
18. Line 294-295, no relationship with the topic.
19. Line 302, Changes of ARGs before and after disinfection, no relation with the ballast water or sediment. Moreover, Chlorination, Ultraviolet, Ozone, and Advanced oxidation were conventional water or wastewater treatment techniques, and many studies have reported its effects on the removal of ARGs, ARB, pathogen, and even the horizontal transfer of resistance genes. There are no new perspectives or ideas in the paper. Besides, the review should focus on the ballast water and sediment, and many technique was not suit for the treatment of ballast water.
20. Line 309-311, rewrite that sentence.
21. Line 340-342, please give the reference.
22. Line 402-406, repeat with the Line 327 and 348.
23. Figure 4 and 5, please give the source of data.
24. Line 419-421, no relation with the section of “Sub-inhibitory concentrations of disinfectants”.
25. Line 448-449, rewrite that sentence.
26. Please double check the format of references.